# Immune Checkpoint Inhibitors and Lupus Erythematosus

**DOI:** 10.3390/ph17020252

**Published:** 2024-02-15

**Authors:** Hans Vitzthum von Eckstaedt, Arohi Singh, Pankti Reid, Kimberly Trotter

**Affiliations:** 1Department of Medicine, University of Chicago Medicine, Chicago, IL 60637, USA; hans.vitzthumvoneckstaedt@uchicagomedicine.org; 2College of the University of Chicago, University of Chicago, Chicago, IL 60637, USA; singharohi@uchicago.edu; 3Committee on Clinical Pharmacology and Pharmacogenomics, University of Chicago Medicine, Chicago, IL 60637, USA; 4Department of Medicine, Section of Rheumatology, University of Chicago Medicine, Chicago, IL 60637, USA; ksmith14@bsd.uchicago.edu

**Keywords:** irAE, lupus erythematosus, immunotherapy, immune checkpoint inhibitors

## Abstract

Immune checkpoint inhibitors (ICIs) are the standard of care for a growing number of malignancies. Unfortunately, they are associated with a broad range of unique toxicities that mimic the presentations of primary autoimmune conditions. These adverse events are termed immune-related adverse events (irAEs), of which ICI-lupus erythematosus (ICI-LE) constitutes a small percentage. Our review aims to describe the available literature on ICI-LE and ICI treatment for patients with pre-existing lupus. Most diagnoses of ICI-LE had findings of only cutaneous lupus; four diagnoses of ICI-LE had systemic lupus manifestations. Over 90% (27 of 29) of cases received anti-PD-1/PDL-1 monotherapy, 1 received combination therapy, and 1 received only anti-CTLA-4 treatment. About three-fourths (22 of 29 or 76%) of patients with ICI-lupus were managed with topical steroids, 13 (45%) received hydroxychloroquine, and 10 (34%) required oral corticosteroids. In our case series, none of the patients with pre-existing lupus receiving ICI therapy for cancer had a flare of their lupus, but few had de novo irAE manifestations, all of which were characterized as low-grade. The review of the literature yielded seven ICI-LE flares from a total of 27 patients with pre-existing lupus who received ICI. Most flares were manageable without need for ICI cessation.

## 1. Introduction

Immune checkpoint inhibitors (ICIs) are immunomodulatory monoclonal antibodies that have revolutionized cancer treatment. Ipilimumab was the first approved ICI in 2011 to treat malignant melanoma [1]. ICIs primarily block the interaction between T lymphocyte checkpoint proteins and their partner proteins on tumor cells. Programmed cell death 1 (PD-1) and cytotoxic T lymphocyte antigen 4 (CTLA-4) are two types of checkpoints expressed on the surface of T cells, functioning as “off” switches to regulate T-cell-mediated immune responses [2]. Checkpoint blockade includes three large classes of medications: PD-1 antibodies (pembrolizumab, nivolumab, dostarlimab, retifanlimab, and cemiplimab), antibodies to the ligand of PD-1 or PDL-1 antibodies (atezolizumab, durvalumab, and avelumab), and CTLA-4 antibodies (ipilimumab and tremelimumab).

While the benefits of harnessing the immune system to attack malignant cells have been demonstrated across numerous trials, a substantial risk exists for a unique group of toxicities referred to as immune-related adverse events (irAEs) [3]. irAEs can affect essentially any organ system in the body. Individual irAE severity is measured by the Common Terminology Criteria for Adverse Events (CTCAE) and is graded from 1 to 5 [4]. Among the less common toxicities are ICI-induced rheumatic conditions, the most common of which are inflammatory arthritis and polymyalgia rheumatica (PMR) [5,6]. This article reviews the current literature on one of the rarer rheumatic irAEs (R-irAEs): ICI-induced lupus erythematosus.

There is a paucity of data on lupus erythematosus (LE) in the setting of checkpoint inhibitor usage. LE is a chronic inflammatory disease characterized by autoantibodies, immune complex (IC) deposition, and immune dysregulation, and it can impact multiple organ systems [7]. Treatment of LE involves a range of different immunosuppressant medications, often guided by the organ systems affected [7]. Here, we present a review of ICI-associated LE. The majority of published accounts are limited to small case reports and case series. The following terms were used to identify PubMed and Google Scholar publications: checkpoint and systemic lupus erythematosus, checkpoint and cutaneous lupus erythematosus. We examined the associations between immune checkpoints and lupus erythematosus, focusing on de novo lupus associated with ICI usage. We further summarize the data available for lupus patients receiving ICIs and whether it influences the risk for disease flares. Finally, we present a small case series of LE patients at our own institution who received immunotherapy between 2011 and the present day.

## 2. Role of Checkpoints in the Pathogenesis of Lupus Erythematosus

### 2.1. PD-1 and PDL-1 and Lupus Erythematosus

PD-1 and its ligands (PDL-1 and PDL-2) are immune regulatory molecules that have been implicated in the pathogenesis of LE, contributing to the loss of immune tolerance and the development of autoimmunity. Expression of these proteins is dysregulated in lupus, though the exact repercussions of this dysregulation remain undetermined [8,9]. Functioning as membrane-bound proteins, normal interactions between PD-1 and its ligands lead to attenuation of the adaptive immune response and allow for self-tolerance [2,10]. As tumor cells express PDL-1 and PDL-2 to avoid the immune system, this pathway emerged as a prime target for immunotherapy [2]. Soluble forms of these proteins are also referred to as sPD-1, sPDL-1, and sPDL-2 [11]. These soluble forms are believed to interact across the PD1 axis by binding to the membrane-bound PD1, PDL-1, and PL-2 and may additionally be linked to lupus [11].

Low levels of sPDL-2 and elevated expression of PDL-2 have been associated with SLE disease activity [11]. In particular, patients with SLE suffering from arthralgias, kidney disease, oral ulcers, and hypocomplementemia have significantly lower sPDL-2 than patients without such manifestations [11]. Murine models have also demonstrated a role for the PD-1 axis in SLE. For example, mice lacking PD-1 expression develop spontaneous lupus-like autoimmune arthritis and glomerulonephritis accompanied by IgG3 and C3 deposition, suggesting loss of peripheral self-tolerance [12]. An additional mouse study found that injecting PDL-1 immunoglobulin into SLE-prone mice demonstrated a protective function, with reduced proteinuria, decreased production of abnormal cytokine levels, and lower anti-dsDNA antibodies [13]. In turn, SLE patients would be expected to express reduced levels of PD-1 and PDL-1. However, contrary to this, PD-1 and PDL-1 expression on immune cells are significantly higher in SLE patients when compared to healthy donors [14,15]. One proposed explanation for these findings is that a defect in tolerance mediated by the PD-1 axis, rather than a lack of PD-1, defines the roles of PD-1s and their ligands in SLE [15]. This hypothesis is bolstered by a study examining the serum levels of autoantibodies against PD-1, which found that elevated levels of anti-PD-1 IgG correlated with increased disease activity and T-cell proliferation in new SLE patients [16]. The PDCD1 gene, a member of the immunoglobulin gene superfamily, codes for PD-1 and is thought to play a direct role in self-tolerance, the suppression of autoreactive B lymphocytes, and immunity as polymorphisms involving the gene are associated with an increased relative risk for SLE [17,18]. Finally, two other pathways that play significant roles in regulating the PD-1 axis, type-I IFN and the Toll-like receptor (TLR) pathways, are also highly active in the pathophysiology of LE [19,20,21,22,23]. Overall, there is evidence that the PD-1 axis influences lupus. However, the exact role of PD-1 and its ligands (both membrane-bound and soluble) in LE and the implications of this relationship on therapy are areas that need further investigation.

### 2.2. CTLA-4 and Lupus Erythematosus

CTLA-4 is a T-cell-specific protein receptor implicated in lupus development and activity. Expressed constitutively on regulatory T-cells (Tregs) and upregulated in conventional T-cells following activation, it plays an important role in immune tolerance through several mechanisms [24]. The exact function of CTLA-4 in lupus development and activity is unclear; however, genetic, serological, and mouse studies provide evidence for its role in autoimmune diseases. Genome-wide studies originally detected CTLA-4 as a potential susceptibility gene for SLE [25]. These reports, in turn, led to investigations of the CTLA-4 gene locus, with multiple studies finding associations between the CTLA-4 gene and promoter region polymorphisms and the development of SLE [26,27].

Furthermore, CTLA-4 knockout mice develop a dysregulated T-cell immune response, resulting in autoimmune disease and severe lupus-like syndrome [28]. An additional murine model examined the effects of blocking CTLA-4′s costimulatory protein CD28, which led to the prevention of lupus nephritis development, prolonged animal survival, and reduced production of dsDNA antibodies [29]. Finally, multiple studies suggest that abnormal expression and function of CTLA-4 contribute to the amplified conventional T-cell responses seen in SLE patients and contribute to the onset and progression of disease [30,31,32]. These investigations have, in turn, led to clinical trials investigating the efficacy of abatacept, a CTLA-4 analog, in SLE treatment, though without great efficacy [33].

## 3. Immune Checkpoint Inhibitor-Associated Lupus Erythematosus

R-irAEs are particularly rare, though likely underreported, with a prevalence of approximately 3–10% and accounting for <1% of all irAEs [34,35,36]. Asymmetric polyarthritis is the most common R-irAE [6,34]. Despite their low incidence, rheumatic events tend to persist for longer periods of time despite cessation of therapy [37]. The development of lupus erythematosus after ICI therapy (ICI-LE), whether it be skin-limited (ICI-associated cutaneous lupus erythematosus (ICI-CLE)) or systemic (ICI-associated systemic lupus erythematosus (ICI-SLE)), is rarely reported. The majority of the literature available is confined to case reports and small case series, the preponderance of which describes CLE. Details on these patients are found in Table 1 [38,39,40,41,42,43,44,45,46,47,48,49,50,51,52,53,54]. Patients rarely met criteria for SLE [55]. In total, 29 cases of ICI-LE were identified in the literature. A total of 27 (93.2%) cases were associated with anti-PD-1 or PDL-1 inhibitor treatment, 1 (3.4%) with CTLA4i treatment, and 1 (3.4%) with combination ICI therapy, in this case ipilimumab and nivolumab (Table 2). Fifteen (51.7%) of all patients were female, while fourteen (48.3%) were male (Table 2). Table 2 shows the associated malignancies of each patient. A total of 9 (31%) patients had melanoma, 7 (24.1%) had non-small-cell lung cancer (NSCLC), and 13 (44.8%) had other cancers. Other malignancies included ovarian carcinoma, breast adenocarcinoma, squamous cell carcinoma, diffuse large B-cell carcinoma, small-cell lung cancer, epidermoid carcinoma, and hepatocellular carcinoma. The onset of ICI-LE was, on average, 6.14 months after ICI initiation, while the average time to ICI-LE resolution was 2.1 months, with 2/29 (6.8%) of cases having refractory symptoms (Table 2). ICI therapy was halted in 15/29 (51.7%) of patients permanently, while 7/29 (24.1%) resumed therapy, 3/29 (10.3%) continued therapy despite ICI-LE development, 3/29 (10.3%) did not have available data if ICI was continued or discontinued, and 1/29 (3.4%) switched ICIs (Table 2). Autoantibody presence varied across cases and, in some cases, was not reported (Table 1).

### 3.1. ICI-CLE

Among the patients afflicted with CLE, subacute CLE (SCLE) is the predominant diagnosis, with 25 out of 29 cases in total (Table 2). All cases of SCLE in the literature were supported by biopsy. Two cases of chilblain lupus are identified in the literature [38,39]. Both patients received pembrolizumab and were clinically diagnosed rather than receiving biopsies (Table 1). A single case of de novo discoid lupus erythematosus (DLE) is reported in the literature (Table 2) [40].

### 3.2. ICI-SLE

From a SLE standpoint, four cases are reported in the literature (Table 2). Three of the four cases were in the setting of anti-PD-1 therapy, and manifestations also included CLE [38,39,52]. The other SLE case was attributed to the anti-CTLA-4 antibody ipilimumab. This is the single case of anti-CTLA-4 SLE in the literature and the only case of lupus nephritis identified [41]. Diagnosis was confirmed in the setting of proteinuria, positive autoantibodies, and suggestive biopsy findings (Table 1) [41].

### 3.3. Management of ICI-LE

The treatment of ICI-associated lupus erythematosus (ICI-LE) varied across cases and is summarized in Table 1. Hydroxychloroquine (HCQ), oral and topical steroids were the most common treatment regimens. A total of 22/29 (75.8%) patients received topical steroids, 13/29 (44.8%) received HCQ, and 10/29 (34.4%) received oral steroids. There was one report of treatment with quinacrine amongst the cases [42]. In another single report, infliximab plus prednisone was initiated, but symptoms did not resolve until infliximab was discontinued and topical corticosteroids were added [43]. In one case, no treatment beyond halting pembrolizumab was required [44].

## 4. Pre-Existing Lupus and Treatment with Immune Checkpoint Inhibitors

### Case Series of Patients with Lupus Receiving Immunotherapy

To expand the available literature on patients with lupus receiving cancer immunotherapy, we queried our institution’s electronic medical record (EMR). We conducted a retrospective case series of patients who had a diagnosis of lupus erythematosus and then subsequently received ICI therapy for their cancer at The University of Chicago Medical Center between January 2011 and July 2023. The study was approved by the Institutional Review Board at the University of Chicago Medical Center. Patients included in the study were at least 18 years old, had a confirmed diagnosis of lupus erythematosus by the treating rheumatologist, received ICI therapy, and had follow-up available in the EMR. Patients were excluded if their symptoms did not meet the diagnosis for lupus erythematosus if they did not have a diagnosis of lupus before ICI initiation, or if the patient did not have any follow-up and clinical annotation available within our EMR. We used at least two ICD search codes to identify patients with lupus and ICI treatment receipt. ICI therapy regimens included monotherapy with an anti-CTLA-4 agent, anti-PD-1/PDL-1 therapy, or a combination of both. For the included patients, abstracted data included demographic characteristics, prior history of LE, details of LE diagnosis, information on cancer diagnosis, ICI treatment regimen and development of pre-existing autoimmune disease flare and/or de novo irAE(s). ICI toxicities were graded according to the CTCAE rubric. 

Our data are summarized in Table 3 and Table 4 below. A total of six patients were identified that met the criteria. Two (33.3%) of these patients were diagnosed with SLE alone, while one (16.6%) carried a diagnosis of DLE, and three (50%) had SLE with cutaneous manifestations (Table 4). Our patients’ prior and current disease manifestations were generally mild and included arthritis, fatigue, and photosensitivity (Table 3). A single patient had biopsy-proven lupus nephritis over a decade prior to commencing immunotherapy. Of note, at the time of ICI initiation, this patient was not on lupus nephritis treatment as his disease was considered quiescent but had received rituximab in the past as part of chronic lymphoblastic leukemia treatment a number of years prior. At the time of ICI initiation, no patients were reporting severe disease manifestations, and all were on minimal therapy (Table 3). Four of six (66.6%) patients received hydroxychloroquine alone, one received topical desonide cream, and one was on no therapy at all. No patients had changes to their LE regimen after initiating their respective cancer therapy. The mean age at LE diagnosis was 49.5 years, though two patients did not have such data available, while the mean age at cancer diagnosis was 60.83 years (Table 4). Malignancies included NSCLC in 4/6 (66.6%) patients, SCC in one patient, and a neuroendocrine tumor in one patient. Four patients (66.6%) were female, while two patients (33.3%) were male (Table 4). As reported in Table 4, all six patients received immunotherapy with either a PD-1 or PDL-1 targeted therapy and received concurrent chemotherapy. All concurrent chemotherapy included a platinum-based component. Three patients (50%) developed a total of five de novo irAEs, none of which constituted flares of their lupus (Table 4). IrAEs included vitiligo, thrombocytopenia, myocarditis, colitis, and arthritis. Four (80%) of the reported irAEs were CTCAE grade 1–2, while there was a single grade 3 irAE in a patient diagnosed with ICI myocarditis. Of note, this patient was the individual with a prior history of more extensive disease manifestations, including lupus nephritis. Treatment of above- mentioned irAEs all included at least temporary discontinuation of ICI (Table 3). Two patients required permanent discontinuation of immunotherapy due to their toxicities (Table 4). The first of these patients was our individual with myocarditis, while the second was the patient suffering from grade 2 thrombocytopenia (Table 3). All other patients either continued to undergo treatment with their ICI or completed their full therapy course. Table 3 reports individual toxicity treatments, which included topical steroids (for vitiligo), oral steroids, intravenous steroids, leflunomide, mesalamine, and vedolizumab. The latter two treatments were for ICI colitis. Three (60%) of reported irAEs, including ICI-myocarditis, improved or resolved with treatment, while two (40%) persisted despite treatment (Table 3). Two (33.3%) of the patients had cancer progression, two (33.3%) had stable disease, and two (33.3%) had disease response at the time of our query (Table 4). 

## 5. Review of Pre-Existing Lupus and ICI Treatment

We then looked at pre-existing lupus and the use of ICIs in literature. Current publications are mostly confined to retrospective analyses, and case reports with the single aforementioned prospective study that included one case of CLE. Our findings are summarized in Table 5. There is one report of a severe SLE flare in the setting of pembrolizumab treatment [56]. This patient had a history of severe disease manifestations, including seizures and renal involvement, though at the time, was well controlled with mild arthralgias [56]. There are two reports of DLE in the literature [53,57]. Initially, ICIs had been avoided in both of these DLE patients due to the fear of disease flares. Blakeway et al. reported a case of a 55-year-old woman’s DLE flaring while on anti-PD-1 therapy [53]. Of note, this patient’s DLE had previously required multiple systemic therapies to control [53]. Zakharian and Lee also reported a patient with lupus treated with pembrolizumab with a mild DLE flare [57]. In contrast to the case reported by Blakeway et al., this patient’s DLE had not required immunosuppressive therapy before ICI initiation. Among all studies and case reports of patients with autoimmune diseases on ICIs, a total of 27 patients had lupus [53,56,57,58,59,60,61,62,63,64,65]. Unfortunately, reported demographics, details on disease activity, current lupus therapy, and other characteristics of this group are sparse. A total of seven patients had lupus flares, six of which were graded 1–2 by CTCAE, while one flare was grade 3. Additionally, five de novo irAEs (in five separate patients) occurred in this cohort of 27. Three of these were graded 1–2, while two grade 3–4 irAEs were graded. Of these higher grade irAEs, one patient with DLE developed grade 3 central diabetes insipidus, while a separate patient with SLE was diagnosed with grade 4 immune thrombocytopenia (ITP) [56,58]. Of interest, the individual who developed ITP was also one of the patients with a grade 3 lupus flare, and had a history of severe lupus manifestations, including neurologic involvement, Libman–Sacks endocarditis, and renal involvement, amongst others [56]. Two patients developed both a lupus flare and a new irAE simultaneously [56,60]. All patients who developed irAEs or lupus flares were undergoing treatment with anti-PD1/PDL1 therapy. In total, 21/27 (77.7%) patients were treated with anti-PD1/PDL-1 therapy, while 3/27 (11.1%) patients received ipilimumab, and 3/27 (11.1%) patients did not have the type of immunotherapy reported in the literature. A total of 11/27 (40.7%) patients were on no therapy at the time of ICI initiation, 3/27 (11.1%) were on HCQ alone, while one (3.7%) patient was on HCQ and oral steroids, and one (3.7%) patient was on oral steroids alone. No data regarding LE treatment was available for 11/27 (40.7%) patients. 

When looking at lupus specifically, the literature is limited at this point, and it is difficult to make definitive claims regarding the risk of lupus flares or irAE occurrence. However, the fact that most patients were either on minimal or no therapy suggests that baseline disease activity in these patients was low at ICI initiation. As mentioned above, the only individual with a grade 3 SLE flare had a history of severe disease manifestations, though at the time, had mild disease by SLE Disease Activity Index (SLEDAI) score, and developed a grade 4 irAE in the form of ITP [56]. Whether or not patients with more active disease at baseline or in the past are at a higher risk of lupus flares and more severe irAEs needs further investigation.

## 6. Review of Any Pre-Existing Autoimmune Disease and ICI Treatment

We then searched more broadly in the literature for patients with any pre-existing autoimmune disease who had subsequently received ICI therapy for cancer. Patients with autoimmune diseases have historically been excluded from immunotherapy clinical trials, largely due to concerns that ICIs may exacerbate their established diseases [66]. With the ever-growing usage of ICIs, their connection with R-irAEs, and the increased risk of malignancy in patients with autoimmune conditions, there is substantial concern over whether these therapies can lead to flares of disease and/or increased risk of irAEs [67]. One retrospective study of 462 patients with pre-existing autoimmune diseases (pAIDs) found that patients with pAIDs were at an increased risk for irAEs (HR 1.81, 95% CI 1.21–2.71) and received a higher amount of systemic corticosteroids for irAEs (HR 1.93, 95% CI 1.35–2.76) when compared to those without pAIDs [68]. Another retrospective study comparing 85 patients with pAIDs to 666 without pAIDs further supported these findings, reporting that irAEs of any grade were more frequent with ICI treatment (65.9% vs. 39.9%) [59]. Importantly, these authors found that there was no difference in risk of high-grade ICI toxicities compared to patients without pAIDs [65]. One prospective study on 45 patients with pAIDs found an increased risk of irAEs and shorter irAE-free survival time when compared to 352 patients without pAIDs [64]. Of note, they found no significant difference between overall survival time and malignancy response rate in their population [64]. Overall, the literature suggests that immunotherapy confers an increased risk of disease flares and irAEs in all autoimmune diseases, with the majority of flares being mild in nature.

## 7. Clinical Trial Studying ICI for Patients with pAIDs

There is an ongoing clinical trial for the prospective study of patients with pre-existing autoimmune diseases who require ICI treatment for cancer. This multi-institutional clinical trial, titled AIM-NIVO (Study of Nivolumab in Patients With Autoimmune Disorders and Advanced Malignancies), is a phase 1b study to assess the overall safety of using anti-PD-1 in patients with various autoimmune diseases, including SLE (NCT03816345).

## 8. Discussion

As ICIs become the standard of care for an increasing number of indications, more patients, including those with pre-existing autoimmune diseases, will be at risk for irAEs. Amongst these, ICI-associated lupus erythematosus is an infrequent but emerging entity. The majority of cases appear confined to CLE, with rare presentations of SLE. Of the ICIs, far more de novo cases of lupus are found in patients on therapy influencing the PD-1 axis.

For patients with pre-existing lupus, the literature suggests that there is a risk of lupus flares while on ICIs. While the vast majority of reported flares have been mild, there is a paucity of data on whether higher baseline disease activity or prior severe disease influences the risk of flares. Notably, the one higher grade flare in the literature occurred in a patient with significant prior disease manifestations, who also developed a significant grade 4 irAE [56]. Our case series did not report any disease flares. However, our sample was only six patients. In line with the available literature, most reported irAEs in our patient population were low grade, with a single incidence of a grade 3 irAE. Similar to aforementioned patient in the literature, this individual had a history of more severe LE manifestations in the past, having been diagnosed with Class V lupus nephritis. Further research is necessary to ascertain which individuals with lupus are more at risk for flares and high-grade irAEs with ICI treatment. We recommend assessing patients’ baseline lupus status before starting ICI treatment. This assessment should include a detailed history of their rheumatic symptoms, prior immunosuppressive regimens, and labs, including urinalysis and characteristic autoantibodies.

While treatment for ICI-LE in previously published literature has been limited to topical or systemic steroids and hydroxychloroquine, other potential treatments to consider include mycophenolate mofetil, tacrolimus, or other medications used to treat lupus, which has already been successfully utilized for other irAEs. A well-powered, prospective study with systemic steroids and steroid-sparing immunomodulating agents will be essential to identify effective therapy that does not abrogate ICI’s anti-tumor immunity. Different immunosuppressants have their concerns: systemic glucocorticoids nonspecifically decrease the immune system and may have a negative impact on anti-tumor immunity, whereas TNF inhibitors can be worrisome to use for ICI-LE as they themselves can cause drug-induced lupus.

## 9. Conclusions

Immune checkpoints play a notable but complicated role in lupus pathogenesis. ICI-LE is a rare toxicity of checkpoint inhibitor therapy but may lead to deleterious cessation of an efficacious cancer treatment. Patients with pre-existing LE should not be indiscriminately precluded from ICI treatment as most cases of LE flares and irAEs were mild with successful symptom resolution, often with the resumption of therapy and, in some cases, without the halt of the offending ICI. Future large, centralized databases and standardized reporting systems for both ICI-lupus and patients with pre-existing LE requiring ICIs will allow for a better understanding of which patients are at risk and provide more direction for the care of these patients.

## Figures and Tables

**Table 1 pharmaceuticals-17-00252-t001:** ICI-Induced Lupus Erythematosus. Characteristics of patients diagnosed with immune checkpoint inhibitor-induced lupus erythematosus.

Patient	Age	Sex	History of Autoimmune Disease	Prior irAE	Malignancy	ICI	Time to Lupus Manifestation Onset (Months)	Diagnosis	Systemic Lupus Erythematosus Criteria Met [55]	Positive Serologies	Negative Serologies	Histopathology	Treatment	Outcomes of Lupus Manifestations	Time to Lupus Symptom Resolution	Resumption of ICI?	Tumor Outcome	Reference
1	74	M	None	None	SCLC	Durvalumab	2	SCLE	None	ANA, SSA, SSB	Anti-dsDNA, Anti-Sm	H&E: Superficial perivascular infiltrate, epidermal atrophy with marked interface change, thin and necrosed epidermis with dysmaturation of atypical basal keratinocyes	HCQ, prednisolone, topical corticosteroid	Resolved	1 month	No	Progression	Pratumchart et al., 2022 [45]
2	58	F	AIHA	None	NSCLC	Nivolumab	5	SCLE	None	SSA, anti-cardiolipin	NR	H&E: Epidermal atrophy, an interface dermatitis composed of a lymphocytic and histiocytic infiltrate and moderate basal vacuolar damage with the presence of colloid bodies	HCQ, prednisolone, topical corticosteroid	Resolved	NR	Yes	NR	Liu et al., 2018 [46]
3	62	F	None	None	NSCLC	Pembrolizumab	47	SCLE	None	ANA, SSA, SSB	NR	H&E: Interface dermatitis with Civatte bodies	HCQ, prednisolone, topical corticosteroid	Resolved	3 months	NR	NR	Andersson et al., 2021 [47]
4	80	M	None	None	Melanoma	Pembrolizumab	6.75	SCLE	None	SSA, anti-cardiolipin	Anti-dsDNA	H&E: Infiltration of lymphocytes in the basement membrane zone, the superficial dermis, and perivascular regions	Prednisolone, topical corticosteroid	Resolved	3 months	No	NR	Ogawa et al., 2020 [48]
5	54	M	None	None	Melanoma	Pembrolizumab	7	SCLE	None	NR	NR	H&E: Interface dermatitis, perivascular and perifollicular lymphocytic infiltrate, occasional dyskeratotic keratinocytes	None	Resolved	1 month	No	NR	Shao et al., 2017 [44]
6	75	F	None	None	Serous Ovarian carcinoma	Ipilimumab + Nivolumab	1.5	SCLE	None	ANA, SSA	Anti-dsDNA, Anti-Sm, SSB	H&E: Interface lymphocytic infiltrate and focal basal vacuolar change	HCQ, quinacrine, prednisone, topical corticosteroid	Resolved	2 months	Switched to Pembrolizumab	NR	Kosche et al., 2019 [42]
7	60	M	None	None	SCLC	Nivolumab	0.5	SCLE	None	SSA	NR	H&E: Interface dermatitis	HCQ, prednisone, topical corticosteroid	Resolved	2 months	No	Progression	Marano et al., 2019 [43]
8	60	F	None	None	NSCLC	Pembrolizumab	0.5	SCLE	None	ANA, SSA, SSB, anti-histone	NR	H&E: Interface dermatitis with adnexal involvement and increased dermal mucin	Prednisone, infliximab, topical corticosteroid	Resolved	1 month	No	No Response	Marano et al., 2019 [43]
9	54	F	None	Psoriasis	SCLC	Nivolumab	20	SCLE	None	ANA, SSA, SSB	Anti-dsDNA	H&E: Focal interface dermatitis, focal lichenoid dermal lymphocytes infiltrate, and mild dermal mucin deposition	HCQ, topical corticosteroid	Resolved	6 months	Continued (no interruption)	NR	Bui et al., 2021 [49]
10	54	F	None	None	Ovarian carcinoma	Pembrolizumab	4	SCLE	None	NR	ANA, SSA, SSB, Anti-dsDNA	H&E: Interface dermatitis, epidermal spongiosis, superficial dermal perivascular lymphocytes infiltrate with rare eosinophils, follicular plugging and subtle dermal mucin deposition	Topical corticosteroid	Resolved	2 months	Yes	NR	Bui et al., 2021 [49]
11	57	F	None	Sjogren’s, Colitis	Breast adenocarcinoma	Atezolizumab	11.5	SCLE	None	ANA, SSA	SSB, Anti-dsDNA	H&E: Interface dermatitis, focal lichenoid infiltrate, superficial to mid-dermal perivascular lymphocytic infiltrate, perifollicular plugging and increased dermal mucin deposition	Topical corticosteroid	Resolved	1 month	No	Progression	Bui et al., 2021 [49]
12	65	M	None	None	SCLC	Pembrolizumab	3	SCLE	None	ANA, SSA	SSB, Anti-dsDNA	H&E: Prominent interface dermatitis, focal vesicle formation, lichenoid infiltrate, prominent dyskeratotic keratinocytes with epidermal necrosis, and superficial to mid-dermal perivascular, periadnexal lymphocytic infiltrate and follicular plugging	HCQ, topical corticosteroid	Resolved	2 months	No	Progression	Bui et al., 2021 [49]
13	60	M	None	None	Melanoma	Nivolumab	0.5	SCLE	None	ANA, SSA	SSB	H&E: Prominent interface dermatitis, lichenoid infiltrate, clefting, prominent superficial to deep dermal perivascular, periadnexal lymphocytic infiltrate and increased dermal mucin deposition	Topical corticosteroid	Resolved	2 months	Continued (no interruption)	NR	Bui et al., 2021 [49]
14	75	M	None	None	NSCLC	Nivolumab	3.75	SCLE	None	ANA, SSA	NR	H&E: Lymphoid inflammatory infiltrate in the upper dermis with moderate basal vacuolar damage and an appreciable dermal mucin deposit with thickening of basement membrane	Prednisone	Resolved	NR	Yes	NR	Diago et al., 2021 [50]
15	66	F	None	None	NSCLC	Nivolumab	6.75	SCLE	None	ANA, SSA	NR	H&E: Lymphoid inflammatory infiltrate in the upper dermis with moderate basal vacuolar damage, and an appreciable dermal mucin deposit with thickening of basement membrane	Prednisone	Refractory	NA	No	NR	Diago et al., 2021 [50]
16	49	FF	None	None	Oropharyngeal SCC	Pembrolizumab	0.5	Chilblain CLE, SLE	SLICC: chronic cutaneous lupus (chilblain lupus), lymphopenia, positive anti-nuclear antibody, positive anti-Smith antibody and low C3	ANA, SSA, Anti-Sm	SSB, Anti-dsDNA, Anti-phospholipid, ANCA, Cryglobulins	NA	HCQ, prednisolone	Resolved	NR	NR	NR	Takeda et al., 2021 [39]
17	52	M	None	None	NSCLC	Pembrolizumab	1.5	SCLE	None	SSA	NR	H&E: Focal vaculoar interface dermatiti, perivascular lymphocytic infiltrate	Prednisolone	Resolved	NR	NR	NR	Gambicher et al., 2021 [51]
18	48	F	None	None	Breast adenocarcinoma	Atezolizumab	1.5	SCLE	None	NR	ANA, anti-dsDNA, ENA	H&E: Inflammatory monomorphic lymphocyte infiltrate in perivascular and periadnexal sites throughout the dermis	Topical corticosteroid	Resolved	0.5 months	Continued (no interruption)	Partial Response	Michot et al., 2019 [38]
19	80	F	None	None	DLBCL	Nivolumab	3.5	SCLE	None	NR	ANA, anti-dsDNA, ENA	H&E: Inflammatory perivascular lymphocytic infiltrate of the upper and middle dermis	Topical corticosteroid	Resolved	0.75 months	No	Progression	Michot et al., 2019 [38]
20	66	F	None	None	Epidermoid carcinoma	Nivolumab	1	SCLE	None	ANA, SSA, ENA (SSA)	Anti-dsDNA	H&E: Perivascular lymphocytic infiltrate of the upper dermis with discrete vacuolization of the epidermal basal layer	Topical corticosteroid	Resolved	0.5 months	Yes	Progression-free survival	Michot et al., 2019 [38]
21	63	M	None	None	Melanoma	Pembrolizumab	5.5	SCLE, SLE	SLICC: SCLE, arthralgia, positive antibodies	ANA, SSA, SSB, ENA (SSA, SSB)	Anti-dsDNA	H&E: Lichenoid dermatosis with staged apoptotic bodies in the epidermis. Peripheral inflammatory mononuclear infiltrate in the upper dermis	HCQ, topical corticosteroid	Resolved	1 month	No	CR	Michot et al., 2019 [38]
22	48	M	None	None	Melanoma	Pembrolizumab	2.5	Chilblain CLE	None	NR	ANA, anti-dsDNA	NA	Topical corticosteroid	Resolved	0.5 months	No	Partial Response	Michot et al., 2019 [38]
23	61	M	None	None	HCC	Nivolumab	21	DLE	None	ANA 1:80	Unreported, 2 months following treatment SSA and anti-histone negative	H&E: Lichenoid interface inflammation with numerous dyskeratotic keratinocytes, pigment incontinence, parakeratosis, follicular plugging, and a dermal perivascular lymphocytic infiltrate	HCQ, topical corticosteroid	Resolved	2 months	Yes	Progression-free survival	Marjunath et al., 2022 [40]
24	64	M	None	None	Melanoma	Ipilimumab	1.5	Lupus nephritis	EULAR 2019: lupus nephritis, antibodies	ANA, anti-dsDNA (regressed following ipilimumab halt)	NR	Kidney bx: Hypertrophy of podocytes and extramembranous deposits. An immunofluorescence study revealed extramembranous and mesangial deposits of IgG, IgM, C3, and C1q. Electron microscopy confirmed the presence of granular, electron-dense extramembranous deposits.	Prednisone	Resolved	12 months	No	CR	Fadel et al., 2009 [41]
25	52	F	None	None	NSCLC	Pembrolizumab	0.5	SLE, SCLE	EULAR 1997: SCLE, arthritis, antibodies	ANA	NR	H&E: Epidermal mild atrophy, vacuolization of epidermal keratinocytes, perivascular inflammatory cell infiltration and epidermis leukocytes exocytosis	HCQ, prednisone	Refractory	NA	No	NR	Ceccarelli et al., 2021 [52]
26	79	F	None	None	Melanoma	Pembrolizumab	2.5	SCLE	None	None	ANA	H&E: Vacuolar interface dermatitis with colloid bodies and dermal perivascular lymphocytic infiltrate	Topical corticosteroid	Resolved	0.75 months	Yes	Partial Response	Blakeway et al., 2019 [53]
27	75	M	None	None	Melanoma	Pembrolizumab	4.5	SCLE	None	None	ANA	H&E: Vacuolar interface dermatitis with colloid bodies and dermal perivascular lymphocytic infiltrate	Topical corticosteroid	Resolved	0.75 months	Yes	Partial Response	Blakeway et al., 2019 [53]
28	72	F	None	Hepatitis	Melanoma	Nivolumab	11	SCLE	None	ANA, SSA, SSB	dsDNA	H&E: Lymphoid inflammatory infiltrates predominantly in perivascular areas, with focal lesions of the dermis and epidermis	HCQ, topical corticosteroid	Resolved	3 months	No	CR	Zitouni et al., 2019 [54]
29	43	M	None	None	NSCLC	Nivolumab	1.5	SCLE	None	ANA, SSA	NR	H&E: Lymphoid perivascular inflammatory infiltrates	HCQ, prednisone, topical corticosteroid	Resolved	0.5 months	No	Progression	Zitouni et al., 2019 [54]

Legend: AIHA—Autoimmune Hemolytic Anemia; aPD-1—Anti-programmed cell death protein 1; ANA—Anti-nuclear antibody; dsDNA—double-stranded DNA; NR—Not Reported; NA—Not Applicable; HCQ—Hydroxychloroquine; CR—Complete Response; H&E—Hematoxylin and Eosin; EULAR—European Alliance of Associations for Rheumatology; SLICC—Systemic Lupus International Collaborating Clinics; irAE—Immune-related adverse event; ICI—Immune Checkpoint Inhibitor; NSCLC—Non-small-cell lung cancer; SCLC—Small cell lung cancer; SCC—Squamous cell carcinoma; DLBCL—Diffuse large B-cell lymphoma; SLE—Systemic Lupus Erythematosus; SCLE—Subacute Cutaneous Lupus Erythematosus; CLE—Cutaneous Lupus Erythematosus; DLE—Discoid Lupus Erythematosus.

**Table 2 pharmaceuticals-17-00252-t002:** Descriptive Data ICI-Induced Lupus Erythematosus. Descriptive data of the patients with ICI-induced Lupus Erythematosus.

Variable	Value
Mean Age (Years)	62.28
Sex	15/29 (51.7%) F, 14/29 (48.3%) M
History of Autoimmune Disease	1/29 (3.4%)
Prior irAE	3/29 (10.3%)
Malignancy	9/29 (31%) Melanoma, 7/29 (24.1%) NSCLC, 13/29 (44.8%) Other
ICI	27/29 (93.2%) aPD-1/aPDL-1, 1/29 (3.4%) anti-CTLA-4, 1/29 (3.4%) Combination
Mean time to lupus manifestation onset (months)	6.14
Diagnosis	4/29 (13.7%) SLE diagnosis, 25/29 (86.2%) SCLE diagnosis, 2/29 (8.6%) Chilblain diagnosis, 1/29 (3.4%) DLE diagnosis
Systemic Lupus Erythematosus criteria met [55]	4/29 (13.7%) met criteria
Outcomes of lupus manifestations	27/29 (93.2%) with resolution, 2/29 (6.8%) with refractory manifestations
Time to lupus symptom resolution	2.1
Resumption of ICI?	7/29 (24.1%) resumed ICI, 3/29 (10.3%) continued ICI through manifestations, 3/29 (10.3%) did not have data reported, 15/29 (51.7%) halted ICI permanently, 1/29 (3.4%) switched ICIs

Legend: aPD-1—Anti-programmed cell death protein 1; aPDL-1—Anti-programmed death ligand 1; CTLA-4—Cytotoxic T-lymphocyte-associated protein 4; ICI—Immune Checkpoint Inhibitor; NSCLC—Non-small-cell lung cancer; SLE—Systemic Lupus Erythematosus; SCLE—Subacute Cutaneous Lupus Erythematosus; CLE—Cutaneous Lupus Erythematosus; DLE—Discoid Lupus Erythematosus.

**Table 3 pharmaceuticals-17-00252-t003:** Patients with Lupus Erythematosus on Checkpoint Inhibitors. Case series patients from the University of Chicago on checkpoint inhibitors.

Pt ID	Sex	Age at Lupus Diagnosis	Lupus Type	Therapy (Prior to ICI)	Age at Cancer Diagnosis	Malignancy	Prior LE Manifestations	Antibodies	ICI Regimen	ICI Start Date	Concurrent Chemotherapy	Changes to LE Tx at ICI Start	LE Tx after ICI Start	Toxicity Type	CTCAE	Toxicity Tx	Adverse Effects of Toxicity Tx	Toxicity Outcome	Cancer Progression after ICI?	ICI Tx Status
1	F	NR	SLE + CLE	HCQ	69	NSCLC	Sicca symptoms, photosensitive rash, Raynaud’s	dsDNA+, RNP+	Durvalumab; Atezolizumab	5/12/2023, then 6/21/23	Cisplatin + Pemetrexed; Abraxane	None	HCQ	NA	NA	NA	NA	NA	PD	Ongoing
2	F	43	SLE + CLE	HCQ	65	Neuroendocrine tumor	Arthralgias, rash	dsDNA+, SSA+	Atezolizumab	12/24/2019	Carboplatin + Etoposide	None	HCQ	Vitiligo, Thrombocytopenia	Grade 1; Grade 1	Desonide	None	Ongoing; Resolved	Stable	Discontinued
3	M	63	DLE	None	62	NSCLC	Rash	ANA+	Durvalumab	11/2/2022	Cisplatin + Pemetrexed	None	Desonide cream	NA	NA	NA	NA	NA	Stable	Ongoing
4	M	42	SLE	Lose dose prednisone, HCQ	59	NCSLC	Inflammatory arthritis, arthralgias and fatigue, lupus nephritis (Class V)	ANA+, dsDNA+, RNP+, Sm+, SSB+	Pembrolizumab	11/17/2020	Cisplatin + Pemetrexed	None	HCQ	Myocarditis	Grade 3	IV solumedrol, prednisone	NR	Resolved	PD (deceased)	Discontinued
5	F	50	CLE + SLE	Hydroxychloroquine	55	NSCLC	Arthralgias, oral ulcers, photosensitivity	ANA+, dsDNA+	Pembrolizumab	2/19/2019	Cisplatin + Pemetrexed	None	Hydroxychloroquine	Colitis, OA =/− IA	Grade 2; Grade 2	Prednisone, Mesalamine, Vedolizumab, Leflunomide	None	Resolved; Ongoing	CR	Held, resumed, completed
6	F	NR	SLE	HCQ	55	Nasopharyngeal SCC	NR	NR	Pembrolizumab	10/7/2022	Carboplatin + Gemcitabine	None	None	NA	NA	NA	NA	NA	PR	Ongoing

Legend: LE—lupus erythematosus; M—male; F—female; NSCLC—non-small-cell lung cancer; SCC—squamous cell carcinoma; SLE—systemic lupus erythematosus; DLE—discoid lupus erythematosus; CLE—cutaneous lupus erythematosus; irAE—immune-related adverse event; ICI—immune checkpoint inhibitor; HCQ—hydroxychloroquine; OA—osteoarthritis; IA—inflammatory arthritis; CR—complete response; PD—progressive disease; PR—partial response; NR—not reported; NA—not applicable; Tx—treatment.

**Table 4 pharmaceuticals-17-00252-t004:** Descriptive Data of Lupus Patients on Checkpoint Inhibitors. Descriptive data of University of Chicago patients with lupus erythematosus who received checkpoint inhibitors.

Patient Information	
Variable	n (%) of Total Six Patients
Median Age LE Diagnosis in years (IQR)	60.5 (56, 64.3)
Sex n (%)	M = 2 (33.3%) F = 4 (66.6%)
Malignancy Type	
NSCLC	4 (66.6%)
SCC	1 (16.6%)
Neuroendocrine	1 (16.6%)
Median Age Malignancy Diagnosis in years (IQR)	46.5 (42.8, 53.3)
Lupus Type	
SLE	2 (33.3%)
DLE	1 (16.6%)
SLE + CLE	3 (50%)
Immunotherapy Type	
Anti- PD-1/PDL-1	6 (100%)
Concurrent Chemotherapy	6 (100%)
Patients with irAEs	3 (50%)
# Lupus Flares	0
IrAE Information	
Total # irAEs	n (%) of Total Five irAEs
CTCAE grading	
Grade 1–2 irAEs	4 (80%)
Grade 3–4 irAEs	1 (20%)
irAE Outcomes	
Resolved/Improved	3 (60%)
Ongoing	2 (40%)
ICI and Cancer Outcomes of Note for Six Total Patients	
Permanent Discontinuation of ICI	2 (33%)
Malignancy Outcome	
Progression	2 (33%)
Stable	2 (33%)
Response	2 (33%)

Legend: aPD-1—Anti-programmed cell death protein 1; aPDL-1—Anti-programmed death ligand 1; LE—lupus erythematosus; IQR—interquartile range; M—male; F—female; NSCLC—non- small-cell lung cancer; SCC—squamous cell carcinoma; SLE—systemic lupus erythematosus; DLE—discoid lupus erythematosus; CLE—cutaneous lupus erythematosus; irAE—immune-related adverse event; ICI—immune checkpoint inhibitor; #—number.

**Table 5 pharmaceuticals-17-00252-t005:** Literature Review Lupus Erythematosus Patients on Checkpoint Inhibitors. Data on lupus erythematosus patients obtained through a literature review.

Patient	Age	Sex	Malignancy	ICI	Lupus Type	Prior Lupus Manifestations	irAE Type	Lupus Tx at Time of ICI Start	Lupus Activity at Time of ICI Start (SLEDAI if Reported for SLE Patients)	Time to Flare/irAE (Months)	CTCAE Grade (Lupus Flare)	CTCAE Grade (irAE)	Tx of Flare/irAE	Outcomes of Lupus Flare/irAE	Time to Lupus Flare/irAE Resolution (Months)	ICI Resumption?	Tumor Outcome	Reference
1	53	F	Melanoma	Pembrolizumab	DLE	Discoid rash	NA	HCQ	NR	0.75	Grade 1–2	NA	Systemic corticosteroids	NR	NR	No	Progression	Blakeway et al., [53]
2	66	F	NSCLC	Pembrolizumab	SLE	Inflammatory arthritis, thrombocytopenia, AKI, Libman–Sacks endocarditis, and generalized seizures	ITP	Low dose prednisone	Active, mild (SLEDAI 4)	0.5	Grade 3	Grade 4	Methylprednisolone, HCQ, IVIG	Resolved	NR	No	Stable	Spagnoletti et al., [56]
3	NR	NR	Melanoma	Anti-PD-1	SLE	NR	Unknown	HCQ	Active, unspecified	NR	Grade 1–2	Grade 1–2	None	NR	NR	Continued	NR	Tison et al., [60]
4	NR	NR	Melanoma	Anti-PD-1	SLE	NR	NA	HCQ	Active, unspecified	NA	NA	NA	NA	NA	NA	NA	NR	Tison et al., [60]
5	NR	NR	Melanoma	Ipilimumab	SLE	NR	NA	None	Inactive	NA	NA	NA	NA	NA	NA	NA	NR	Tison et al., [60]
6	NR	NR	Melanoma	Anti-PD-1	SLE	NR	NA	None	Inactive	NA	NA	NA	NA	NA	NA	NA	NR	Tison et al., [60]
7	NR	NR	NSCLC	Anti-PD-1	SLE	NR	NA	None	Inactive	NA	NA	NA	NA	NA	NA	NA	NR	Tison et al., [60]
8	NR	NR	NSCLC	Anti-PD-1	SLE	NR	Vitiligo	None	Inactive	NR	Grade 1–2	Grade 1–2	None	NA	NA	NA	NR	Tison et al., [60]
9	NR	NR	NSCLC	Anti-PD-1	SLE	NR	Vitiligo	None	Inactive	NR	Grade 1–2	Grade 1–2	None	NA	NA	NA	NR	Tison et al., [60]
10	68	M	NSCLC	Pembrolizumab	SLE	Discoid rash	NA	NR	NR	NR	NA	Grade 3	Topical corticosteroids	Resolved	NR	Continued	Partial response	Zakharian et al., [57]
11	NR	NR	NR	Anti-PD-1	SLE	NR	NA	NR	Inactive	8.5	Grade 1–2	NA	Topical corticosteroids, prednisone	Improved	1	Yes	Stable	Leonardi et al., [58]
12	NR	NR	NR	Anti-PD-1	SLE	NR	Central DI	NR	Inactive	2.5	NA	NA	Desmopressin	Resolved	NR	Yes	NR	Leonardi et al., [58]
13	NR	NR	UC	Anti-PD-1	SLE	NR	NA	None	Inactive	5	NA	NA	Systemic corticosteroids	Improved	NR	Yes	NR	Martinez Chanza et al., [59]
14	NR	NR	UC	Anti-PD-1	SLE	NR	NA	None	Active, mild	NA	NA	NA	NA	NA	NA	NA	NR	Martinez Chanza et al., [59]
15	NR	NR	UC	Anti-PD-1	SLE	NR	NA	None	Inactive	NA	NA	NA	NA	NA	NA	NA	NR	Martinez Chanza et al., [59]
16	NR	NR	UC	Anti-PD-1	SLE	NR	NA	None	Active, mild	NA	NA	NA	NA	NA	NA	NA	NR	Martinez Chanza et al., [59]
17	NR	NR	UC	NR	DLE	NR	NA	NR	NR	NR	Grade 1–2	NA	NR	NR	NR	NR	NR	Martinez Chanza et al., [59]
18	NR	NR	UC	NR	DLE	NR	NA	NR	NR	NA	NA	NA	NA	NA	NA	NA	NR	Martinez Chanza et al., [59]
19	NR	NR	RCC	NR	DLE	NR	NA	NR	NR	NA	NA	NA	NA	NA	NA	NA	NR	Martinez Chanza et al., [59]
20	NR	NR	Melanoma	Ipilimumab	SLE	Arthralgias	NA	Prednisone, HCQ	NR	NA	NA	NA	NA	NA	NA	NA	NR	Johnson et al., [62]
21	NR	NR	Melanoma	Ipilimumab	SLE	NR	NA	HCQ	NR	NA	NA	NA	NA	NA	NA	NA	NR	Johnson et al., [62]
22	NR	NR	Melanoma	Anti-PD-1	SLE	NR	NA	NR	Inactive	NA	NA	NA	NA	NA	NA	NA	NR	Menzies et al., [61]
23	NR	NR	Melanoma	Anti-PD-1	SLE	NR	NA	NR	Inactive	NA	NA	NA	NA	NA	NA	NA	NR	Menzies et al., [61]
24	NR	NR	NR	Anti-PD-1	CLE	NR	NA	NR	NR	NA	NA	NA	NA	NA	NA	NA	NR	Danlos et al., [64]
25	NR	NR	NR	Anti-PD-1	DLE	NR	NA	NR	NR	NA	NA	NA	NA	NA	NA	NA	NR	Kaur et al., [63]
26	NR	NR	NR	Anti-PD-1	SLE	NR	NA	None	Inactive	NA	NA	NA	NA	NA	NA	NA	NR	Cortellini et al., [65]
27	NR	NR	NR	Anti-PD-1	SLE	NR	NA	None	Inactive	NA	NA	NA	NA	NA	NA	NA	NR	Cortellini et al., [65]

Legend: aPD-1—Anti-programmed cell death protein 1; ANA—Anti-nuclear antibody; dsDNA—double-stranded DNA; NR—Not Reported; NA—Not Applicable; HCQ—Hydroxychloroquine; CR—Complete Response; H&E—Hematoxylin and Eosin; EULAR—European Alliance of Associations for Rheumatology; SLICC—Systemic Lupus International Collaborating Clinics; SLEDAI—Systemic Lupus Erythematosus Disease Index; irAE—Immune-related adverse event; ICI—Immune Checkpoint Inhibitor; ITP—Immune Thrombocytopenia; NSCLC—Non-small-cell lung cancer; UC—Urothelial Carcinoma; RCC—Renal cell carcinoma; SLE—Systemic Lupus Erythematosus; SCLE—Subacute Cutaneous Lupus Erythematosus; CLE—Cutaneous Lupus Erythematosus; DLE—Discoid Lupus Erythematosus; Tx—Treatment.

## Data Availability

All data is available within this manuscript in the provided tables.

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
