# Peer review of "Immune Checkpoint Inhibitors and Lupus Erythematosus"

_pharmaceuticals, 2024, doi:10.3390/ph17020252_

Round 1

Reviewer 1 Report

Comments and Suggestions for Authors

 Immune Checkpoint Inhibitors and Lupus Erythematosus

This manuscript discussed an important topic. However, some important point need to be revised

15.. with few reports of systemic lupus…like what part?

16…...on PD-1/PDL-1 therapy. ....... It is necessary to follow the scientific rules in writing abbreviations.

It is better to rewrite the results in the abstract section so that they contain numbers.

32.. There is a lot of information and multiple sentences without adding the main reference to them.

42.. checkpoint 42 inhibitor....... It is necessary to follow the scientific rules in writing abbreviations.

68.. oral ulcer ...language editing and grammar revision are mandatory.

Table 1. Unreadable table

Table 2. The table is not easy to interpret.

Table 3. unreadable table

Also, the arrangement of table 4 is not easy to understand.

Comments on the Quality of English Language

Moderate editing of English language required

Author Response

Prof. Dr. Amelia Pilar Rauter                                                                          January 15, 2024

Editor-in-Chief

Pharmaceuticals

Dear Professor Doctor Rauter,

Thank you for your email enclosing the reviewers’ comments regarding our manuscript entitled “Immune checkpoint inhibitors and Lupus erythematosus.” The comments were highly insightful and allowed us to improve the quality of our manuscript. We have carefully reviewed the comments and accordingly revised our manuscript. Our responses in blue below are provided in a point-by-point manner:

REVIEWER 1: This manuscript discussed an important topic. However, some important point need to be revised:

  • . with few reports of systemic lupus…like what part?

Author reply: To clarify for the reviewer, we edit’ed this sentence to the following: “Most diagnoses of ICI-lupus had only findings of cutaneous lupus; few diagnoses of ICI-lupus had systemic lupus manifestations.”  

  • 16…...on PD-1/PDL-1 therapy. .......It is necessary to follow the scientific rules in writing abbreviations.

Author reply: Thank you. We revised this sentence to the following: “Most cases had received anti-PD-1/PDL-1 therapy and we found only a single case who had received anti-CTLA-4 treatment.”

  • It is better to rewrite the results in the abstract section so that they contain numbers.

Author reply: Thank you for this recommendation. We have edit’ed the abstract to include specific numerical results. The abstract now reads, “Immune checkpoint inhibitors (ICIs) are the standard of care for a growing number of malignancies. Unfortunately, they are associated with a broad range of unique toxicities that mimic the presentations of primary autoimmune conditions. These adverse events are termed immune-Related Adverse Events (irAEs), of which ICI-lupus constitutes a small percentage. Our review aims to describe the available literature on ICI-lupus and the treatment of lupus patients with ICIs by searching PubMed and Google Scholar. Most diagnoses of ICI-lupus had findings of only cutaneous lupus; four diagnoses of ICI-lupus had systemic lupus manifestations.  Over 90% (27 of 29) of cases had received anti-PD-1/PDL-1 monotherapy, 1 who had received combination therapy and 1 case who had received only anti-CTLA-4 treatment. About three-fourths (22 of 29 or 76%) of patients with ICI-lupus were managed with topical steroids, 13 received hydroxychloroqurine and 10 (34%) required oral corticosteroids. In our cases series of patients with pre-existing lupus receiving ICI therapy for cancer, none had flare of their lupus diagnoses, but few had de novo irAE manifestations, all of which were characterized as low-grade. Review of literature yielded 7 ICI-lupus flares from total of 27 patients with pre-existing lupus who had received ICI. Most flares were able to be managed. “

  • .There is a lot of information and multiple sentences without adding the main reference to them. 

Author reply: We have provided necessary references. The following sentences is solely a list of most commonly used checkpoint inhibitors at the time of our submission: “Checkpoint blockade includes three large classes of medications: PD-1 antibodies (pembrolizumab, nivolumab, dostarlimab, retifanlimab, and cemiplimab), antibodies to ligand of PD-1 or PDL-1 antibodies (atezolizumab, durvalumab, and avelumab), and CTLA-4 antibodies (ipilimumab and tremelimumab).” This sentences was formulated based on the authors’ delineating the names of the checkpoint inhibitors rather than getting information for this sentence from any one particular source.

  • .checkpoint 42 inhibitor....... It is necessary to follow the scientific rules in writing abbreviations.

Author reply: Thank you for this – we will make sure to apply the proofreading option from Microsoft word to address any errors from a grammatical standpoint.

  • .oral ulcer ...language editing and grammar revision are mandatory. 

Author reply: Thank you for this – we will make sure to apply the proofreading option from Microsoft word to address any errors from a grammatical standpoint.

  • Table 1. Unreadable table

Author reply: Separate document with the table has been provided in the submission.

  • Table 2. The table is not easy to interpret.

Author reply: Separate document with the table has been provided in the submission.

  • Table 3. unreadable table

Author reply: Separate document with the table has been provided in the submission.

  • Also, the arrangement of table 4 is not easy to understand.

Author reply: Separate document with the table has been provided in the submission.

The authors provide novel data along with a comprehensive literature review on a very significant topic. There are, however, multiple potential issues to be addressed:

  • the authors report on six patients with lupus before ICI. Disease information about these patients is scarce. 
    1. Did these patients meet the 1997, 2012 or 2019 classification criteria?
    2. How active were they at time of ICI start? Was the SLEDAI calculated?
    3. How significant their damage burden (e.g. measured through the SLICC/ACR damage index) was at time of ICI start?
    4. Which treatment did these patients receive from disease onset?

Author reply: Not all of these patients had systemic lupus erythematosus. The characteristics of their disease is delineated in our Table 3. Unfortunately, we only had information that was available in the clinical annotations and therefore not all of the details regarding their disease activity.

  • The authors also report that no patient in their series had a lupus flare. However, a case of thrombocytopenia and one of myocarditis are reported. Both manifestations can be part of the clinical spectrum of SLE? How did the authors differentiate between "pure" ICI-related manifestations and ICI-induced lupus manifestations in this setting? Of note, thrombocytopenia was also reported in other studies (Table 5)

Author reply: Thank you for asking about this clarification. Patients were deemed to have a flare of their pre-existing lupus if the manifestations of their ICI-related toxicity was a worsening of their baseline manifestations. Additionally, if the patient did not have other findings of their lupus active (ie rash, arthritis, etc.), the primary caretakers in the clinical notes had characterized these findings (thrombocytopenia or myocarditis) as de novo irAE instead of flare of their lupus.

  • Aggregate data (Table 2 and 4) are very interesting to comprehensively understand the relationship between SLE and ICI. However, formal and content issues exist in this context:
    1. analyzing the authors' patients together with those reported in the literature (current Table 5), would be more interesting that only referring to the former alone
    2. both table 3 and 4 are bad structured and look more like notes: it would be useful to better separate distinct contents
    3. median (interquartile range) instead of mean would be preferred, given the low number of subjects

Author reply:

  • In terms of comparing our series with the rest of the literature: We wanted to caution before comparing the two populations, as our numbers are quite small. Additionally, we wanted to avoid presenting our entire patient panel alongside those published as there will be publication bias within the cases accepted through peer-review.
  • Tables 3 and 4: Have been updated so that they are not notes but rather helpful tables to follow. Table 4 has distinct content compared to Table 3.
  • Table 4 has had edits to reflect median and IQRs instead of means. Thank you for this suggestion.

  • other minor issues:
    1. hydroxychloroquine (or HCQ) instead of Plaquenil would be preferred (e.g. in Table 3)
    2. the last two paragraphs before the Discussion ("review of pre-existing autoimmune disease and ICR treatment" and "clinical trial studying ICI for patients with pAID") would be better integrated within the discussion

Author reply: We’ve changed the Plaquenil to hydroxychloroquine. We have changed around our paper to expanded on critical components. We hope that the discussion flows a bit better now. Thank you so much for your valuable recommendations.

We hope that the revised version and accompanying responses will be sufficient to make our manuscript suitable for publication. We look forward to hearing from you at your earliest convenience.

Sincerely,

Hans Vitzthum von Eckstaedt, MD
and
Pankti Reid, MD MPH
The University of Chicago Medical Center

Reviewer 2 Report

Comments and Suggestions for Authors

The authors provide novel data along with a comprehensive literature review on a very significant topic. There are, however, multiple potential issues to be addressed:

1) the authors report on six patients with lupus before ICI. Disease information about these patients is scarce. 

- Did these patients meet the 1997, 2012 or 2019 classification criteria?

- How active were they at time of ICI start? Was the SLEDAI calculated?

- How significant their damage burden (e.g. measured through the SLICC/ACR damage index) was at time of ICI start?

- Which treatment did these patients receive from disease onset?

2) The authors also report that no patient in their series had a lupus flare. However, a case of thrombocytopenia and one of myocarditis are reported. Both manifestations can be part of the clinical spectrum of SLE? How did the authors differentiate between "pure" ICI-related manifestations and ICI-induced lupus manifestations in this setting? Of note, thrombocytopenia was also reported in other studies (Table 5)

3) aggregate data (Table 2 and 4) are very interesting to comprehensively understand the relationship between SLE and ICI. However, formal and content issues exist in this context:

- analysing the authors' patients toghether with those reported in the literature (current Table 5), would be more interesting that only referring to the former alone

- both table 3 and 4 are bad structured and look more like notes: it would be useful to better separate distinct contents

- median (interquartile range) instead of mean would be preferred, given the low number of subjects

4) other minor issues:

- hydroxychloroquine (or HCQ) instead of Plaquenil would be preferred (e.g. in Table 3)

- the last two paragaphs before the Discussion ("review of pre-existing autoimmune disease and ICR treatment" and "clinical trial studying ICI for patients with pAID") would be better integrated within the discussion

Author Response

(The authors gave the same response as above.)
